# A Retrospective Assessment of Neuropathic Pain in Response to Intraneural Facilitation^®^ Therapy and Neurovascular Index-Guided Food Elimination

**DOI:** 10.3390/biomedicines13030688

**Published:** 2025-03-11

**Authors:** Mark Bussell, Kyan Sahba, Hailey Jahromi, Mitra Rashidian, Jamie Hankins

**Affiliations:** 1Neuropathic Therapy Center, Loma Linda University Health, Loma Linda, CA 92354, USA; mbussell@llu.edu (M.B.); hjahromi@llu.edu (H.J.); 2Department of Allied Health Studies, Loma Linda University, Loma Linda, CA 92354, USA; mitrarashidian@gmail.com; 3School of Nursing, Loma Linda University, Loma Linda, CA 92354, USA; jhankins@llu.edu

**Keywords:** neuropathy, pain, INF^®^ Therapy, intraneural facilitation^®^, physical therapy

## Abstract

**Background/Objectives**: To evaluate the effectiveness of our dual approach in treating neural ischemia. **Methods**: Researchers were able to retrospectively audit patient data collected from January 2022–September 2024. Patients were included if they received intraneural facilitation^®^ (INF^®^), participated in neurovascular index (NVI)-guided food elimination, and completed pre and post pain-quality assessment scale (PQAS) forms in its entirety. **Results**: Eighteen of the twenty PQAS descriptive pain variables were significantly different pre- vs. post treatment: intense (*p* = 0.000), sharp (*p* = 0.002), hot (*p* = 0.020), dull (*p* = 0.022), cold (*p* = 0.005), sensitive (*p* = 0.000), shooting (*p* = 0.000), numb (*p* = 0.000), electrical (*p* = 0.000), tingling (*p* = 0.000), cramping (*p* = 0.000), radiating (*p* = 0.000), throbbing (*p* = 0.000), aching (*p* = 0.000), heavy (*p* = 0.000), unpleasant (*p* = 0.000), deep pain (*p* = 0.000), and intense surface pain (*p* = 0.000). Itchy (*p* = 0.058) and tender (*p* = 0.062) were not found to be significant. There was also significance in pain decrease in the three mean domains: paroxysmal (*p* = 0.000), superficial (*p* = 0.000), and deep (*p* = 0.000). **Conclusions**: This study suggests that blending a mechanical intervention (INF^®^) with a lifestyle modification (NVI-guided food elimination) is effective in improving PQAS scores in patients with peripheral neuropathy, indicating a possible reversal of neural ischemia and maintenance of capillary patency.

## 1. Introduction

Peripheral neuropathy (PN) is one of the most common disorders bringing patients into a neuromuscular clinic, commonly with the chief complaint of foot numbness and pain [1]. Peripheral neuropathies in the early stages may present with progressing sensory loss, including numbness, burning sensations, and pain [2]. Later, PN symptoms include proximal numbness, distal weakness, and atrophy [2]. Other commonly experienced PN symptoms are stabbing and electrical pain, allodynia, hyperalgesia, and hyperesthesia [2]. While PN is one of the most common diagnoses, the treatment methods are very ambivalent even though it is a marker for increased risk of death among U.S adults [1,3]. The prevalence of PN in the general population ranges from 1 to 7%, with increased rates in those over 50 years old [2]. The most common cause of PN is from diabetes mellitus (DM), diabetic peripheral neuropathy (DPN), affecting 25–50% of those with DM [2]. The U.S. healthcare estimates a cost of USD 10.9 billion per year for the care of DPN alone [4]. Idiopathic neuropathy, now known as chronic idiopathic axonal polyneuropathy (CIAP) [5], is found in 25–46% of all PN cases, and prevalence rate increases as age advances [2], affecting about 5–8 million Americans [5]. Chemotherapy-induced peripheral neuropathy (CIPN) affects 30–40% of those treated with chemotherapeutic agents [6]. While PNs have heterogeneous presentations with many etiologies [7], there remains one consistent factor, namely neurovascular ischemia [8].

“In human neuropathy, occlusion of the capillaries that supply the nerves causes ischemic nerve fiber damage and perineural capillary luminal occlusion that is due to both endothelial cell hypertrophy and hyperplasia” [9]. For DPN, reductions in blood flow lead to nerve hypoxia; this hypoxia induces the expression of multiple pro-angiogenic and pro-inflammatory genes in macrophages [9]. These microvascular changes lead to a reduced oxygen supply and glucose both from the reduced blood flow and reduced uptake by capillary dysfunction [10]. Similarly, in CIPN, there is an increased presentation of pro-inflammatory cytokines and macrophages, in conjunction with chemotherapy drugs reducing the presence of fenestrated capillaries leading to neuropathy [11]. CIAP sufferers strongly mimic DPN sufferers, as they both have similar basal lamina area thickness, which suggests ischemia plays a role in its neuropathy pathway [12]. They also share other similarities, from capillary amount to number of endothelial cell nuclei [13]. The aforementioned neuropathy pathophysiology commonality seen above in DPN, CIPN, and CIAP is not just ischemia, but capillary dysfunction.

Capillary disturbances for flow and function rather than ischemia may be the mechanism responsible for neuropathy [10]. Common interventions to treat endoneurial capillary closure can be categorized into two categories: lifestyle modifications and mechanical intervention. Lifestyle modifications used for diabetic neuropathy management include Hemoglobin A1C (HbA1C), food modification, and exercise [14,15,16]. Exercise studies are starting to show prevention/improvement in diabetic neuropathy [10,14,15]. Studies reviewing diet modification in conjunction with exercise also show decreased neuropathic pain and increased cutaneous nerve regenerative capacity [7,14,15]. Normalizing HbA1C in type II DM improved microvascular neuropathy complications more effectively than glycemic control [16]. Mechanical interventions involve a direct effect on the nerve, such as acupuncture, pulsed electromagnetic field (PEMF) therapy, low-level-light therapy (LLLT), and electrical stimulation [17,18,19,20]. Acupuncture needles may exert direct force on underlying nerve and peri-neural tissues through manual manipulation or electric current, and studies show their effectiveness in DPN [17]. PEMF therapy is emerging as an effective mechanical intervention for pain reduction in those with DPN and may show improvement in the vascular physiology in microcirculatory dysfunction [20]. Lastly, LLLT, in laboratory studies, is effective in improving neuronal repair, suggesting LLLT as a treatment for neuropathy [19]. LLLT is identified as a complimentary therapy in the treatment of PN and DPN [19]. Additionally, electrical stimulation has been shown to facilitate nerve regeneration and improve microvascular function, which are both critical in maintaining endoneurial capillaries [18]. While these mechanical interventions are accepted for symptomatic relief and functional improvement, there is still a lack of a curative neuropathic treatment [21]. We believe the management of peripheral neuropathy requires a multifaceted approach that includes lifestyle modifications to control blood glucose levels and mechanical interventions to enhance blood flow and address the underlying endoneurial capillary change.

This retrospective audit examines a two-pronged approach: (1) lifestyle wellness through NVI-guided food elimination and (2) addressing endothelial dysfunction via INF^®^ Therapy. NVI is a patented computer assessment of ultrasound measures that makes recommendations based on ultrasound data brought into the NVI software 73r from data gleaned from pre- and post-suspected food allergen ingestion. The NVI assesses waveform and volume-flow changes after the food allergen is ingested and then grades the response. The participating patient will abstain from foods that are given a “Moderate grade” or higher during their therapy, or beyond, depending on the patient’s tolerance of the dietary change. The mechanical intervention used in this study was INF^®^ Therapy, which is a manual therapy treatment provided by a trained licensed physical therapist which aims to guide blood flow into the neural fascicle, improve endoneurial capillary circulation, and ultimately reverse ischemia [21]. INF^®^ Therapy has demonstrated improved neural performance in multiple studies, potentially due to targeting and overcoming microvascular resistance through a series of holds thought to recruit more consistently into the neural fascicle through a series of protracted holds [21,22,23]. The Neuropathic Therapy Center (NTC) almost exclusively treats patients with neuropathic pain or paresthesia. Using retrospective data will allow the NTC to evaluate the treatment response of blending INF^®^ Therapy and lifestyle medicine in our specific patient population and observe pain responses through the pain-quality assessment scale (PQAS).

## 2. Materials and Methods

This retrospective study was conducted at the Neuropathic Therapy Center at Loma Linda University Health in Loma Linda, CA. The study was conducted in accordance with the Declaration of Helsinki (1975, revised in 2013), and approved by the Institutional Review Board at Loma Linda University (IRB#: 5220267). Demographics and information from patient care, including results from procedures, medical records, and clinical questionnaires were collected during routine patient visits from the NTC of Loma Linda University database. The NTC has 2 outpatient clinics in Loma Linda and Murrieta, California and contains a multicenter patient database. Records for all patients who received INF^®^ Therapy during an almost 4-year period (October 2020–September 2024) were included in this database; however, only patients from January 2022–September 2024 were used in this study due to our updated waveform analysis software. Patients were included in the study if they had completed one series of INF^®^ Therapy treatments prior to September 2024. If patients completed more than one series of INF^®^ Therapy treatment, only data from the first treatment was used. All patients regardless of neuropathy severity, or type of neuropathy, comorbidities, or length of symptomatic suffering were considered. Patients were excluded if they received only orthopedic treatment, were currently receiving their first INF^®^ Therapy treatment, or had incomplete/missing data. Patients included in our study were asked to continue their daily lives as normal (meaning continue with their prescribed medication, and lifestyle). Patients that began our program were given 2–4 NVI tests to assess for possible food allergens and were asked to abstain from these foods. We did not monitor their compliance with this. Post-NVI assessments, patients were given INF^®^ Therapy treatments within a timeframe ranging from 4 to 13 visits. Once INF^®^ Therapy treatments were completed, patients were given another PQAS form to complete to assess changes in their neuropathic pain.

### 2.1. Demographic Information

Data collected included demographic variables such as age, gender, race, and information from all routine (standard care) visits including, but not limited to, the following: the results from the PQAS clinical questionnaire, NVI for waveform analysis, medical history and medical records for type of neuropathy, body mass index (BMI), days lapsed between pre and post assessments, number of INF^®^ Therapy treatments.

### 2.2. Pain-Quality Assessment Survey (PQAS)

The PQAS is a numerical pain-rating scale distinguishing between 20 pain descriptors [24]. Subjects read the introduction of the questionnaire and then rated both spatial and quality pain categories on a numeric scale of 0 = ‘no pain’ or ‘no painful sensation’ to 10 = ‘worst imaginable pain sensation’ [24]. From these data, three pain-quality domains were also calculated; paroxysmal (averaging scores of shooting, sharp, electric, hot and radiating); superficial (averaging scores of itching, cold, numb, sensitive and tingling); and deep (averaging scores of aching, heavy, dull, cramping and throbbing) [25].

### 2.3. NVI-Guided Food Testing

Ultrasound software 73r analyzes vascular change in the patients both pre- and post- ingestion of suspected food allergen. The NVI-guided food testing involves a baseline assessment of volume flow and waveforms in ten different areas on the left side of the body. This includes four areas on the left upper extremity and six areas on the lower extremity. The measurements of the waveforms, including both anterograde and retrograde volume flow, as well as pulsatility, are processed using a series of mathematical formulas, resulting in a numerical value that serves as a waveform descriptor. After these initial assessments, the total volume flow of the eight measurements is calculated. The patient then ingests suspected food allergens that are part of their diet. Following this, the initial measurements are repeated to determine any differences between pre- and post ingestion. A rating of the changes is conducted, providing an outcomes-based assessment that indicates whether there was a significant reaction to consuming the suspected allergen. Based on these results, the therapist will recommend whether the specific protein or food should be included in the patient’s diet during therapy. Patients measured with “moderate” severity or greater were asked to abstain from the food group from their diet for the remainder of treatment sessions, and after, if conducive with their lifestyle. The foods primarily tested were gluten, dairy, and sugar; a few were also tested after ingesting nuts, oats, soy, chicken, and beef. The foods tested were determined based on the subjects’ diet and what they mostly ate.

### 2.4. INF^®^ Therapy

INF^®^ Therapy is a new manual modality initiated in South Carolina and brought to Loma Linda University in 2011. INF^®^ Therapy involves a series of manual holds administered by a trained practitioner, which proposes accomplishing three simultaneous events that reestablish circulation to the ischemic nerve [21,22,23]. The first hold optimizes the nerve–artery relationship by placing and holding a joint in its maximal loose-packed position [21] (Appendix A). This stretch bridges nutrient vessels at the facilitated joint, bringing blood flow into epineural arterioles from large, accompanying vessels [21]. The entire neurovascular circulatory system is a closed dual-chamber system that is fed by nutrient vessels from larger arteries into outer-chamber epineural arterioles [21]. This hold pressurizes the outer chamber for the entire neurovascular system, providing available microcirculation for access by the second hold [21]. The second hold is a stretch on the targeted treatment area, enabling pressurized outer-chamber circulation to be released into the inner neurovascular chamber through non-occluded transperineural vessels [21,22] (Appendix A). These vessels feed previously ischemic endoneurial capillaries that are adjacent to the nerve axon by hydraulically restoring blood flow inside the fascicle [21]. Distal to the point of previously described circulatory induction, pressurized microvascular circulation will encounter upstream closed capillary beds, and a third hold is necessary to accelerate endoneurial blood flow while avoiding edema and vascular stasis [21]. The third hold involves bolsters and pressure points away from the secondary hold, promoting distant circulatory flow and creating a type of microvascular vacuum [21,22] (Appendix A). The vacuum would assist with bringing microvascular circulation more readily into connection with patent vascular beds [21]. The third hold completes the three-hold sequence with all three holds combining to establish an endoneurial and epineurial vascular pathway to the ischemic nerve [21,22].

### 2.5. Statistical Analysis

The clinical director ensured the data collected aligns with the principles and policies of NTC’s clinic and Loma Linda University Health. The data were downloaded from the RedCap database, saved on a secured server, and transferred to the statistician through LLUH’s secure email service. The data were cleaned by research staff prior to transmission to the statistician. Data were analyzed using IBM SPSS Statistics (Version 27). Mean ± SD was computed for quantitative variables and frequency (percentage) for ordinal variables. A power analysis was performed for a sample size of 73 subjects, giving us an effect size of 0.85 at a level of significance α = 0.05. The paired *t*-test was used to compare pre- and post variables within the group, as most cases showed data were distributed normally. The Wilcoxon Signed-Rank test was also carried out; the results were not different in terms of significance. Pearson correlation analysis was used to look at the association between PQAS data and NVI data. Non-parametric analysis was used for non-normal data. All analyses were performed at an alpha level of 0.05. The NTC’s clinical director and research staff were with the statistician to interpret and draw conclusions from the data.

## 3. Results

Out of the 156 patients who attended our clinic and completed food testing and our PQAS form, 11 patients were excluded for not having any PQAS data, 37 were excluded for having no post-treatment PQAS data, 4 were excluded for having no pre-treatment PQAS data, and 28 were excluded for not completing their PQAS forms and having missing entries (Figure 1). In total, 73 patients successfully completed the NTC treatment with the survey and data collected from January 2022–September 2024 (Figure 1). Neuropathy diagnoses in this study were: 67.1% CIAP, 8.2%% CIPN, 12.3% DPN, and 12.3% other (post surgical, trauma, neuralgia, ciguatera, causalgia). There were no significant differences between groups except for the large sample size of the idiopathic group (Table 1).

Eighteen of the twenty PQAS descriptive pain variables were significant pre- vs. post treatment: intense (*p* = 0.000), sharp (*p* = 0.002), hot (*p* = 0.020), dull (*p* = 0.022), cold (*p* = 0.005), sensitive (*p* = 0.000), shooting (*p* = 0.000), numb (*p* = 0.000), electrical (*p* = 0.000), tingling (*p* = 0.000), cramping (*p* = 0.000), radiating (*p* = 0.000), throbbing (*p* = 0.000), aching (*p* = 0.000), heavy (*p* = 0.000), unpleasant (*p* = 0.000), deep pain (*p* = 0.000), and intense surface pain (*p* = 0.000). Itchy (*p* = 0.058) and tender (*p* = 0.062) were not found to be significant but were trending towards significance (Table 2). There was also significance in the pain decrease in the three mean domains: paroxysmal (*p* = 0.000), superficial (*p* = 0.000), and deep (*p* = 0.000) (Table 2).

This population (*n* = 73) was split into four neuropathic categories (idiopathic, chemotherapy, diabetic, and other). The CIAP group (*n* = 49) PQAS responses showed the same significance in the above general population (Table 3). Due to the large sample size of idiopathic patients, an independent T test was run. The other categories had a much smaller sample size; therefore, Wilcoxon was used. The chemotherapy or CIPN group (*n* = 6) PQAS responses showed significance only in numb (*p* = 0.027), tingling (*p* = 0.046) and superficial domain (*p* = 0.012) (Table 3). For the diabetic or DIPN group (*n* = 9) PQAS responses showed significance in the sharp (*p* = 0.048), sensitive (*p* = 0.042), tender (*p* = 0.021), shooting (*p* = 0.048), electrical (*p* = 0.015), tingling (*p* = 0.028), cramping (*p* = 0.045), radiating (*p* = 0.017), aching (*p* = 0.027), unpleasant (*p* = 0.050), deep (*p* = 0.027) and paroxysmal domains (*p* = 0.033) (Table 3). For the other group of neuropathy-presenting patients (*n* = 9), PQAS responses showed significance only in the shooting (*p* = 0.045), sharp (*p* = 0.050), electrical (*p* = 0.027), numb (*p* = 0.049), cramping (*p* = 0.017), unpleasant (*p* = 0.027), intense surface (*p* = 0.048), and deep pain domains (*p* = 0.011) (Table 3).

The median number of food tests completed per patient was 3. The foods tested were primarily gluten (100%), dairy (95.9%) and sugar (84.9%), with a few participants who were tested for nuts, oats, soy, chicken, and beef. There are no data for whether patients-maintained diet modification or not. The median number of INF^®^ Therapy sessions completed per patient was 7.

## 4. Discussion

The PQAS has been used to evaluate neuropathic pain associated with Morton’s neuroma, CIPN, and pregabalin treatment [24,26,27]. It is considered to be a more generic instrument which can differentiate between more nociceptive and neuropathic pain conditions [28]. Our study utilized the PQAS for patients feeling any neuropathic pain not defined by a specific disease or disorder. In a previous study by Sahba et al., the PQAS was used and found that the unpleasant pain domain was only found to be significant pre- and post INF^®^ Therapy, where 8 out of the 20 pain-quality domains were found significantly improved (*p* ≤ 0.05), for 17 subjects [21]. The pain-quality factors that were found to be significant were also found in the small-nerve-fiber symptoms, involving multiple symptoms such as burning, shooting, and numbness. This can indicate that INF^®^ Therapy can affect small nerve fibers and alleviate conditions that restrict endoneurial blood flow. In this study, 18 out of the 20 domains were found significant, and although this study had a larger sample size (*n* = 73), eliminating high-inflammatory foods was not utilized in the Sahba et al. study. However, the Sahba et al. study illuminated a placebo effect in both the INF and sham groups that they found to be consistent with the formation of a ‘sustained partnership’ between the healthcare provider and patient [21]. Thus, we believe that further research should be carried out, exploring not only the importance of lifestyle modification in conjunction with INF^®^ Therapy, but also including the impact that the provider–patient relationship can have on pain symptoms.

The PQAS contains 20 domains in total: 2 global (pain intensity and unpleasantness), 2 spatial (deep and surface), and 16 quality domains (sharp, hot, dull, cold, sensitive, tender, itchy, shooting, numb, electrical, tingling, cramping, radiating, throbbing, aching and heavy) [25]. The 16 quality domains can be further categorized into 3 pain-quality factors: (1) paroxysmal pain sensations (shooting, sharp, electric, hot and radiating), (2) superficial pain (itchy, cold, numb, sensitive, and tingling), and (3) deep pain (aching, heavy, dull, cramping, and throbbing) [25]. The pain-quality domain not included in the grouping is tender, as it did not strongly correlate with any of the above [25]. All three pain-quality factors were significantly improved (*p* ≤ 0.05) when compared pre- vs. post treatment, as well as every single pain-quality domain (*p* ≤ 0.05). These results share similarities with the study produced by Alshahrani et al., where significance was seen between pre and post treatment measurements of the modified Total Neuropathy Scale, Sensory Organization Test, and the limits of stability test. These tests were important to test whether this therapy decreased symptoms of neuropathy, improved static balance, dynamic balance, and movement velocity. The overall improvement in pain seen in this pain assessment, along with patient feedback and previous studies generating comparable findings, supports the continuation of this two-pronged approach to treating peripheral neuropathy.

Food intake involving gluten regardless of gluten-dependent or non-gluten-dependent related chronic diseases still has pro-inflammatory side effects, inducing organ dysfunction and neurodegenerative changes [29,30]. Gluten sensitivity, as marked by detected antibodies against transglutaminase 6 in the sera of patients, commonly leads to peripheral neuropathy [31,32]. Of those with gluten neuropathy, 55% experience pain, but when adhering to a gluten-free diet, pain decreased, and overall health improved [31,32,33]. Upon ingestion, gluten becomes partially hydrolyzed by proteases in the gastrointestinal tract, becoming peptides [34]. These peptides undergo deamidation by intestinal tissue transglutaminase-2, thus increasing their affinity for major histocompatibility complex II molecules and prompting an inflammatory response [34]. For those with gluten sensitivity, without celiac disease, it is suspected that the gluten proteins in relation to other wheat components activate an immune response within the body, increasing pain and inflammatory response [34].

The relationship between gluten, neuropathy, and the possible mechanisms involving epithelial tight junctions and nitric oxide release is complex and multifaceted. Recent studies highlight nitric oxide (an inflammatory mediator) in the pathogenesis of various neuropathies, including DIPN. In diabetic neuropathy, the increase in nitric oxide and other inflammatory mediator syntheses contributes to neuropathic pain [35], thus suggesting that inflammation and oxidative stress play a role in the development of neuropathy, potentially exacerbated by dietary factors such as gluten. Epithelial tight junctions are critical components of the epithelial barrier, regulating permeability and maintaining homeostasis within the intestinal environment [36,37]. In those with gluten sensitivity and celiac disease, ingesting gluten triggers an immune response, disrupting the integrity of the tight junction and resulting in increased intestinal permeability, also known as leaky gut [36,37].

Interventions aimed at restoring tight-junction integrity have been explored as therapeutic strategies for managing gluten-related disorders. Clinical trials using larazotide acetate have reduced the inflammatory process associated with celiac disease and some dietary supplements have been suggested to protect against gluten-mediated tight-junction injury [38,39]. These proposed interventions face significant challenges due to biological complexities, variability in individual responses, practical implementation issues, and regulatory hurdles, thus limiting their effectiveness and safety [36,37,38].

The significance of targeted reinstituting blood flow to neural closed capillary beds cannot be understated. As expressed in the study by Alshahrani et al., during the first hold of INF^®^ Therapy, nutrient vessels are stretched, which subsequently enlarges the opening at the junction of the artery and bridging nutrient vessel [22]. These openings essentially create increased epifascial vascular pressure, facilitating a pressurized blood flow to cross into the endoneurial capillaries. However, due to a strong perineurium and increased endoneurial edema, a secondary hold is necessary to drive blood flow into the endoneurial capillaries. This hold involves a mild stretch to allow for circulation to be driven past the perineurium and into these capillaries. Finally, a third hold is used to create a vacuum within the vasculature to pull microvasculature circulation into the intraneural vascular beds [23]. If this extra-capillary vascular pressure exceeds the resistance of potentially closed neural capillaries, during the three holds of INF^®^ Therapy, then red blood cells may be compressed. Studies show red blood cells that are forced through simulated tight capillary lumens secrete adenoside triphosphate (ATP) [24]. ATP then induces nitric oxide from capillary endothelial cells [25], facilitating endothelial vasodilation. Theoretically, inducing endothelial vasodilation should enhance blood flow in these areas and improve nerve function through the enhancement of the concentration of Na/K ATPase [22]. We hypothesize that, though there are few studies that have investigated this technique, this mechanism of INF^®^ Therapy accounts for the significant changes seen in patient’s PQAS responses before and after the treatment series. Given these results, we still believe that more research is needed to confirm this proposed mechanism, as well as how INF^®^ Therapy can increase neural circulation and alleviate symptoms caused by ischemia-inducing conditions. However, INF^®^ Therapy-induced restoration of circulation to the neural capillary bed is only part of the neuropathy reversal equation. Neural circulatory maintenance extends beyond the initial administration of INF^®^ Therapy and includes inhibiting adverse microvascular factors while promoting factors that favor microvascular health. With vascular stability, neural regeneration will occur [26]. This study suggests the NTC treatment model, blending INF^®^ Therapy (mechanical intervention) with NVI-guided food elimination (lifestyle modification) is potentially effective in improving PQAS scores in patients with peripheral neuropathy, suggesting a possible reverse neural ischemia and maintenance of capillary patency.

### Limitations

This study provides valuable insights into the benefits of using INF^®^ Therapy and NVI-guided food elimination for alleviating neuropathy symptoms. However, several limitations should be considered. As a retrospective analysis of patients seen at the NTC from January 2022 to September 2024, the study is subject to the inherent biases of retrospective designs, such as the inability to randomize participants [40]. Our small sample size is primarily due to our strict inclusion criteria; specifically, not including subjects who left any questions blank. Investigators did not want to risk false data by assuming subjects may have meant 0, or no pain, on the form by not answering. Due to this exclusion criteria alone, 80 subjects were excluded. For our future studies, we hope to emphasize to our subjects that our questionnaire should be filled out in its entirety and emphasize the ability to ask questions if they are unsure, rather than leaving it blank. The sample population was drawn exclusively from the NTC, which limits the generalizability of the findings to the broader population, as it may not fully represent patients from other settings or demographics [41]. Additionally, the absence of a control group further restricts the ability to attribute the improvements in PQAS scores solely to INF^®^ Therapy and lifestyle modifications, as there was no comparison to a placebo or alternative treatment group [42]. Additional limitations were that we had no records for dietary modification compliance and our small sample size due to multiple missing data entries. A further limitation was the potential variability in the physical therapists’ expertise, which could have influenced the delivery and effectiveness of INF^®^ Therapy. The differences in the providers’ skill levels may have contributed to variations in the patients’ reported PQAS scores, highlighting a lack of standardization in treatment delivery. Despite these limitations, this study offers promising findings, and future research could address these issues by incorporating a control group, improving patient randomization, and standardizing the implementation of INF^®^ Therapy, thereby enhancing the reliability and validity of the results.

## 5. Conclusions

Notable improvement in neuropathic symptoms was observed with the application of INF^®^ Therapy in conjunction with NVI-guided food elimination for patients suffering from peripheral neuropathy. Further research, including prospective multi-site studies involving this approach, is recommended.

## Figures and Tables

**Figure 1 biomedicines-13-00688-f001:**
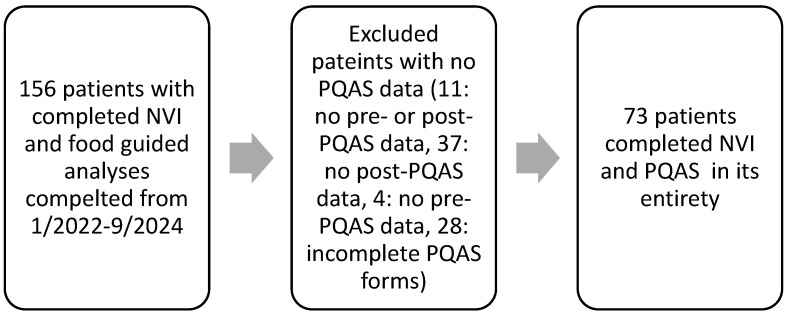
Retrospectively selecting patients.

**Table 1 biomedicines-13-00688-t001:** Demographic variables of subjects.

Variables		Idiopathic	Chemotherapy	Diabetic	Other
**Age**		69.49 ± 10.11 ^a^	65.35 ± 7.93	69.70 ± 4.76	60.31 ± 13.16
**Weight**		81.05 ± 17.29	80.31 ± 15.43	87.11 ± 20.37	82.39 ± 30.94
**BMI**		26.81 ± 5.01	28.25 ± 4.71	29.36 ± 5.61	27.82 ± 11.06
**Number of Treatments**		6.94 ± 2.16	6.50 ± 2.07	6.89 ± 1.90	7.67 ± 2.18
**Number of Food Tests**		3.29 ± 0.61	3.33 ± 0.82	2.78 ± 0.67	3.11 ± 0.78
**Race**	**White**	40 (81.6) ^b^	4 (66.7)	6 (66.7)	5 (55.6)
	**Not-White**	9 (18.4)	2 (33.3)	3 (33.3)	4 (44.4)

^a^: Mean ± SD. ^b^: count (percentage).

**Table 2 biomedicines-13-00688-t002:** Results of the pain quality assessment scale (PQAS) questionnaire pre- and post treatment in patients with peripheral neuropathy that were seen at the Neuropathic Therapy Center (NTC) from January 2022–September 2024 and received intraneural facilitation^®^ (INF^®^) for the treatment of their peripheral neuropathy.

Variables	*n* = 73		95% Confidence Interval of the Difference		*p*-Value	Correlation
	Pre	Post	Lower	Upper		
**Shooting**	3.47 ± 2.94	1.99 ± 2.22	−2.122	−0.837	0.000	0.457
**Sharp**	3.85 ± 3.02	2.78 ± 2.32	−1.738	−0.399	0. 002	0.446
**Electircal**	3.68 ± 3.03	2.03 ± 2.25	−2.447	−0.868	0.000	0.203
**Hot**	4.01 ± 3.11	2.25 ± 2.29	−2.446	−1.089	0.000	0.452
**Radiating**	3.56 ± 2.99	1.97 ± 2.11	−2.275	−0.903	0.000	0.377
**Paroxysmal** **Domain Mean**	3.72 ± 2.25	2.20 ± 1.60	−2.0024	−1.0222	0.000	0.447
**Itchy**	1.42 ± 2.35	1.03 ± 1.84	−0.808	0.014	0.058	0.67
**Cold**	3.08 ± 3.31	2.10 ± 2.24	−1.67	−0.302	0.005	0.496
**Numb**	6.29 ± 3.00	4.27 ± 2.87	−2.68	−1.348	0.000	0.528
**Sensitive**	3.41 ± 3.26	2.19 ± 2.25	−1.823	−0.615	0.000	0.613
**Tingling**	5.47 ± 3.01	3.40 ± 2.52	−2.769	−1.368	0.000	0.422
**Superficial** **Domain Mean**	3.93 ± 1.93	2.60 ± 1.59	−1.7678	−0.9061	0.000	0.463
**Aching**	4.22 ± 3.11	2.42 ± 2.25	−2.491	−1.098	0.000	0.416
**Heavy**	3.81 ± 3.30	2.19 ± 2.22	−2.294	−0.939	0.000	0.504
**Dull**	4.40 ± 2.89	3.42 ± 2.95	−1.801	−0.144	0.022	0.261
**Cramping**	3.95 ± 3.40	2.19 ± 2.43	−2.384	−1.123	0.000	0.616
**Throbbing**	2.59 ± 2.96	1.51 ± 1.92	−1.669	−0.495	0.000	0.539
**Deep** **Domain Mean**	3.79 ± 2.17	2.35 ± 1.66	−1.8741	−1.0136	0.000	0.563
**Intense**	5.64 ± 2.58	3.64 ± 2.10	−2.661	−1.339	0.000	0.283
**Unpleasant**	6.58 ± 2.24	4.22 ± 2.59	−3.021	−1.691	0.000	0.312
**Deep Pain**	5.11 ± 2.99	3.21 ± 2.76	−2.65	−1.158	0.000	0.385
**Tender**	3.40 ± 3.00	2.66 ± 2.51	−1.517	0.037	0.062	0.279
**Intense Surface Pain**	4.97 ± 2.84	3.05 ± 2.42	−2.62	−1.216	0.000	0.355

Data presented as mean ± SD.

**Table 3 biomedicines-13-00688-t003:** Results of the pain-quality assessment scale (PQAS) questionnaire pre- and post treatment in patients grouped by their type of peripheral neuropathy that were seen at the Neuropathic Therapy Center (NTC) from 1/2022 to 9/2024 and received intraneural facilitation^®^ (INF^®^) for the treatment of their peripheral neuropathy.

Idiopathic *n* = 49			95% Confidence Interval of the Difference		*p* Value ^a^	Chemotherapy *n* = 6		95% Confidence Interval of the Difference		*p* Value ^b^	Diabetic *n* = 9		95% Confidence Interval of the Difference		*p* Value ^b^	Other *n* = 9		95% Confidence Interval of the Difference		*p* Value ^b^
PQAS	Pre	Post	Lower	Upper		Pre	Post	Lower	Upper		Pre	Post	Lower	Upper		Pre	Post	Lower	**Upper**	
**Shooting**	3.63 ± 2.86	2.20 ± 2.32	−2.22	−0.64	0.001	2.17 ± 3.92	0.67 ± 0.82	−5.52	2.52	0.382	3.56 ± 3.28	1.33 ± 1.32	−4.42	−0.02	0.048	3.33 ± 2.55	2.33 ± 2.78	−2.58	0.58	0.184
**Sharp**	4.06 ± 3.08	3.06 ± 2.47	−1.83	−0.17	0.019	2.33 ± 3.20	1.00 ± 1.27	−4.56	1.90	0.337	4.11 ± 3.30	1.78 ± 1.09	−4.64	−0.03	0.048	3.44 ± 2.35	3.44 ± 2.24	−1.92	1.92	1.000
**Electrical**	3.73 ± 3.09	2.08 ± 2.08	−2.61	−0.70	0.001	2.67 ± 3.78	2.00 ± 3.52	−6.44	5.11	0.779	4.11 ± 3.14	1.22 ± 1.09	−5.04	−0.73	0.015	3.67 ± 2.35	2.56 ± 3.09	−3.12	0.90	0.239
**Hot**	4.43 ± 3.31	2.37 ± 2.29	−2.96	−1.17	0.000	1.33 ± 1.97	0.50 ± 0.84	−3.35	1.69	0.434	3.67 ± 2.60	2.44 ± 2.65	−2.80	0.35	0.111	3.89 ± 2.32	2.56 ± 2.40	−3.57	0.91	0.207
**Radiating**	3.88 ± 2.88	2.16 ± 2.00	−2.57	−0.86	0.000	1.17 ± 2.04	0.67 ± 0.82	−1.95	0.95	0.415	4.89 ± 3.37	1.33 ± 2.24	−6.28	−0.83	0.017	2.11 ± 2.71	2.44 ± 2.88	−0.68	1.35	0.471
**Paroxysmal Pain** **Domain mean**	3.95 ± 2.16	2.38 ± 1.62	−2.17	−0.98	0.000	1.93 ± 2.54	0.97 ± 1.18	−3.69	1.75	0.403	4.07 ± 2.73	1.62 ± 1.17	−4.36	−0.52	0.019	3.29 ± 1.77	2.67 ± 1.77	−1.53	0.28	0.151
**Itchy**	1.39 ± 2.24	1.02 ± 1.69	−0.86	0.13	0.141	2.17 ± 3.13	2.17 ± 3.06	−0.66	0.66	1.000	1.56 ± 2.92	0.56 ± 0.88	−3.27	1.27	0.340	1.00 ± 2.00	0.78 ± 2.33	−0.86	0.42	0.447
**Cold**	3.43 ± 3.27	2.47 ± 2.36	−1.79	−0.12	0.025	2.00 ± 4.00	1.00 ± 2.45	−2.76	0.76	0.203	2.33 ± 3.87	1.44 ± 1.59	−4.11	2.34	0.543	2.67 ± 2.65	1.44 ± 1.59	−3.28	0.84	0.209
**Numb**	6.08 ± 3.07	4.14 ± 2.87	−2.77	−1.10	0.000	7.50 ± 2.17	4.17 ± 2.32	−5.60	−1.07	0.013	6.78 ± 2.68	5.11 ± 3.02	−4.41	1.08	0.199	6.11 ± 3.55	4.22 ± 3.42	−3.71	−0.07	0.044
**Sensitive**	3.45 ± 3.25	2.27 ± 2.23	−1.99	−0.37	0.005	3.17 ± 3.71	2.17 ± 3.49	−1.94	−0.06	0.041	3.56 ± 3.54	1.44 ± 1.67	−4.12	−0.10	0.042	3.22 ± 3.31	2.56 ± 2.13	−2.20	0.87	0.347
**Tingling**	5.65 ± 3.01	3.61 ± 2.36	−2.90	−1.18	0.000	5.50 ± 3.78	1.67 ± 1.37	−6.90	−0.76	0.024	5.67 ± 2.78	3.22 ± 2.86	−4.55	−0.34	0.028	4.22 ± 2.95	3.56 ± 3.40	−3.10	1.76	0.545
**Superficial Pain** **Domain mean**	4.00 ± 1.85	2.70 ± 1.62	−1.81	−0.79	0.000	4.07 ± 1.81	2.23 ± 1.21	−3.05	−0.62	0.012	3.98 ± 2.38	2.36 ± 1.19	−3.78	0.54	0.122	3.44 ± 2.20	2.51 ± 2.10	−2.19	0.33	0.126
**Aching**	4.45 ± 3.17	2.59 ± 2.21	−2.81	−0.90	0.000	2.50 ± 3.51	1.33 ± 2.42	−3.20	0.87	0.201	4.89 ± 3.14	2.44 ± 2.13	−4.29	−0.60	0.016	3.44 ± 2.35	2.22 ± 2.64	−2.84	0.40	0.120
**Heavy**	4.27 ± 3.32	2.57 ± 2.24	−2.55	−0.83	0.000	1.83 ± 2.04	0.67 ± 0.52	−2.97	0.64	0.158	3.33 ± 3.94	2.00 ± 2.55	−4.16	1.49	0.308	3.11 ± 2.93	1.33 ± 2.06	−3.73	0.17	0.069
**Dull**	4.37 ± 2.82	3.27 ± 2.68	−1.96	−0.24	0.013	4.33 ± 3.72	3.33 ± 3.98	−4.45	2.45	0.490	5.00 ± 3.46	3.56 ± 3.05	−6.35	3.46	0.517	4.00 ± 2.50	4.22 ± 3.90	−2.23	2.68	0.840
**Cramping**	4.10 ± 3.31	2.47 ± 2.54	−2.41	−0.86	0.000	2.00 ± 4.00	0.33 ± 0.52	−6.00	2.67	0.368	3.33 ± 3.54	1.22 ± 1.48	−4.16	−0.06	0.045	5.00 ± 3.35	2.89 ± 2.71	−3.57	−0.65	0.010
**Throbbing**	2.67 ± 3.04	1.67 ± 2.03	−1.76	−0.24	0.011	2.67 ± 3.45	1.17 ± 2.40	−3.96	0.96	0.178	2.78 ± 3.15	0.89 ± 0.93	−4.14	0.37	0.090	1.89 ± 2.37	1.44 ± 1.88	−1.22	0.33	0.225
**Deep Pain** **Domain mean**	3.97 ± 2.17	2.51 ± 1.68	−1.99	−0.92	0.000	2.67 ± 2.43	1.37 ± 1.67	−2.61	0.01	0.051	3.87 ± 2.64	2.02 ± 1.32	−4.02	0.33	0.087	3.49 ± 1.40	2.42 ± 1.87	−1.63	−0.51	0.002
**Intense**	6.08 ± 2.41	3.67 ± 2.09	−3.15	−1.66	0.000	3.33 ± 3.45	2.50 ± 2.88	−4.61	2.95	0.595	5.56 ± 3.05	3.33 ± 2.12	−5.12	0.67	0.115	4.89 ± 1.62	4.56 ± 1.42	−1.96	1.30	0.650
**Unpleasant**	6.82 ± 2.15	4.49 ± 2.60	−3.12	−1.53	0.000	4.50 ± 2.81	2.67 ± 2.50	−5.49	1.83	0.254	6.89 ± 2.42	3.67 ± 2.87	−6.17	−0.28	0.036	6.33 ± 1.73	4.33 ± 2.29	−3.54	−0.46	0.017
**Deep Pain**	5.22 ± 3.05	3.43 ± 2.87	−2.78	−0.81	0.001	2.83 ± 2.48	1.33 ± 1.51	−3.57	0.57	0.122	5.33 ± 3.74	2.00 ± 2.50	−5.88	−0.78	0.017	5.78 ± 1.56	4.44 ± 2.30	−3.09	0.43	0.119
**Tender**	3.45 ± 3.04	2.82 ± 2.56	−1.61	0.34	0.198	1.17 ± 2.04	2.05 ± 3.73	−2.74	5.41	0.438	4.00 ± 3.43	1.67 ± 1.32	−4.22	−0.45	0.021	4.00 ± 2.55	2.89 ± 2.37	−3.40	1.18	0.295
**Intense Surface Pain**	5.29 ± 2.63	3.27 ± 2.40	−2.80	−1.24	0.000	4.00 ± 3.85	2.17 ± 2.99	−5.05	1.38	0.202	3.89 ± 3.48	3.22 ± 2.17	−3.97	2.64	0.654	5.00 ± 2.69	2.33 ± 2.55	−5.13	−0.21	0.037

Data presented as mean ± SD. ^a^: between-group comparison; independent T test. ^b^: between-group comparison; Wilcoxon non-parametric test.

## Data Availability

The raw data supporting the conclusions of this article will be made available by the authors on request.

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
