# Peer review of "A Retrospective Assessment of Neuropathic Pain in Response to Intraneural Facilitation® Therapy and Neurovascular Index-Guided Food Elimination"

_biomedicines, 2025, doi:10.3390/biomedicines13030688_

Round 1

Reviewer 1 Report

Comments and Suggestions for Authors

This study is the result of retrospectively verifying the effect in patients with neuropathic pain in INF Therapy and food allergy tests. The study is well designed, and the small numbers for the verification results are well implemented through verification. First, I would like to point out a few things.

1.  "This retrospective audit used a two-pronged program aimed to address both; 1. Lifestyle wellness, with neurovascular index (NVI), guided food elimination, and 2. Mechanical challenges associated with endothelial dysfunction, with intraneural facilitationTM (INF® therapy)."
Issue: The semicolon after "both" is incorrect. The phrase "aimed to address both" is redundant. Please change this sentence to "This retrospective audit examined a two-pronged approach: (1) lifestyle wellness through NVI-guided food elimination and (2) addressing endothelial dysfunction via INF® Therapy."

2. There is insufficient information on how to do about INF Therapy. Please add a brief description of these and a photo or other supplementary materials

3. "To evaluate our two-pronged treatment response with treating neural ischemia." is unclear. Please change this phrase to ' To evaluate the effectiveness of our dual approach in treating neural ischemia' or similar meanings.

4. Also, describe the food test in detail in terms of materials and methods. It was organized so that you couldn't know what kind of test you did until you read the summary of the results later.

5. The study lacks a control group, making it difficult to determine whether the improvements were due to INF® Therapy or natural pain fluctuations. There is no explicit discussion on whether patients followed their dietary modifications consistently, which may affect results.
   Please clarify how you could deal with this defect. 

6. The authors acknowledge that one author invented INF® Therapy and holds patents on the Neurovascular Index (NVI).
This disclosure is crucial, but I thought additional clarification on how conflicts were described was definitely needed. 

Author Response

REVIEWER 1:

Comment 1.  "This retrospective audit used a two-pronged program aimed to address both; 1. Lifestyle wellness, with neurovascular index (NVI), guided food elimination, and 2. Mechanical challenges associated with endothelial dysfunction, with intraneural facilitationTM (INF® therapy)."
Issue: The semicolon after "both" is incorrect. The phrase "aimed to address both" is redundant. Please change this sentence to "This retrospective audit examined a two-pronged approach: (1) lifestyle wellness through NVI-guided food elimination and (2) addressing endothelial dysfunction via INF® Therapy."

Response 1. Thank you for your knowledgeable and thorough response to our manuscript. We agree. Changes made, we added, “This retrospective audit examined a two-pronged approach: (1) lifestyle wellness through NVI-guided food elimination and (2) addressing endothelial dysfunction via INF® therapy.” (Page 2-3, Introduction paragraph 4, line 90-92)

Comment 2. There is insufficient information on how to do about INF Therapy. Please add a brief description of these and a photo or other supplementary materials

Response 2. We agree with your comment. A certification course is currently being designed for therapists both physical and occupational, to be certified in INF® therapy. There will be multiple modules to complete along with lab and 6 months of clinical practice. It will take over the paper to explain this technique, however we have added more content on what we believe to be the physiology behind it. Supplementary materials will be provided with this response including pictures of the core components of the therapy as described in the following additional therapy description. We added the following “As expressed in the study by Alshahrani et. al., during the first hold of INF® Therapy nutrient vessels are stretched which subsequently enlarges the opening at the junction of the artery and bridging nutrient vessel [22]. These openings essentially create increased epifascial vascular pressure intending for a pressurized blood flow to cross into the endoneurial capillaries. However, due to a strong perineurium and increased endoneurial edema a secondary hold is necessary to drive blood flow into the endoneurial capillaries. This hold involves a mild stretch to allow for circulation to be biased past the perineurium and into these capillaries. Finally, a third hold is used to create a vacuum within the vasculature to pull microvasculature circulation into the intraneural vascular beds [23]. If this extra-capillary vascular pressure, exceeds the resistance of potentially closed neural capillaries, during the three holds of INF® Therapy, then red blood cells may be compressed. Studies show red blood cells that are forced through simulated tight capillary lumens secrete adenoside triphosphate (ATP) [24]. ATP then induces nitric oxide from capillary endothelial cells [25] facilitating endothelial vasodilation. Theoretically, inducing endothelial vasodilation should enhance blood flow in these areas and improve nerve function through the enhancement of the concentration of Na/K ATPase [22]. We hypothesize that, though there are few studies that have investigated this technique, this mechanism of INF® Therapy accounts for the significant changes seen in patient’s PQAS responses before and after treatment series. Given these results, we still believe that more research is needed to confirm this proposed mechanism, as well as how INF® Therapy can increase neural circulation and alleviate symptoms caused by ischemia-inducing conditions. (Page 9-10, Discussion- paragraph 5, line 341-362).

Comment 3. "To evaluate our two-pronged treatment response with treating neural ischemia." is unclear. Please change this phrase to ' To evaluate the effectiveness of our dual approach in treating neural ischemia' or similar meanings.

Response 3. Agreed. We added “To evaluate the effectiveness of our dual approach in treating neural ischemia.” (Page 1, abstract background/objectives section- paragraph 1, line 13-14).

Comment 4. Also, describe the food test in detail in terms of materials and methods. It was organized so that you couldn't know what kind of test you did until you read the summary of the results later.

Response 4. Agreed. We added “The NVI food testing involves a baseline assessment of volume flow and waveforms in ten different areas on the left side of the body. This includes four areas on the left upper extremity and six areas on the lower extremity. The measurements of the waveforms, including both anterograde and retrograde volume flow, as well as pulsatility, are processed using a series of mathematical formulas, resulting in a numerical value that serves as a waveform descriptor. After these initial assessments, the total volume flow of the eight measurements is calculated. The patient then ingests suspected food allergens that are part of their diet. Following this, the initial measurements are repeated to determine any differences between pre- and post-ingestion. A rating of the changes is conducted, providing an outcomes-based assessment that indicates whether there was a significant reaction to consuming the suspected allergen. Based on these results, the therapist will recommend whether the specific protein or food should be included in the patient's diet during therapy. Patients measured with “moderate” severity or greater they were asked to abstain from the food group from their diet for the remainder of treatment sessions, and thus after if conducive with their lifestyle. Foods primarily tested were gluten, dairy and sugar, a few were also tested after ingesting nuts, oats, soy, chicken, and beef. The foods tested were determined based on the subjects’ diet and what they mostly ate.” (Page 4, Materials and Methods- NVI Guided-Food testing- paragraph 2, lines 150-167).

Comment 5. The study lacks a control group, making it difficult to determine whether the improvements were due to INF® Therapy or natural pain fluctuations. There is no explicit discussion on whether patients followed their dietary modifications consistently, which may affect results. Please clarify how you could deal with this defect. 

Response 5. We agree and want to rectify this with future RCT studies. In our limitations paragraph we stated and added the following, “Additionally, the absence of a control group further restricts the ability to attribute the improvements in PQAS scores solely to INF® therapy and lifestyle modifications, as there was no comparison to a placebo or alternative treatment group [1]. Additional limitations were that we had no records for dietary modification compliance and our small sample size due to multiple missing data entries.” (page 10, limitations, line 389-391). We also felt it was necessary to change our original closing discussion statement from “This study suggests the NTC treatment model, blending INF® Therapy (mechanical intervention) with NVI guided food elimination (lifestyle modification) is effective in improving PQAS scores in patients with peripheral neuropathy suggesting a possible reverse neural ischemia and maintenance of capillary patency.” To “is potentially effective in improving PQAS scores” (page 10, discussion- paragraph 1, line 367-370).

Comment 6. The authors acknowledge that one author invented INF® Therapy and holds patents on the Neurovascular Index (NVI).
This disclosure is crucial, but I thought additional clarification on how conflicts were described was definitely needed. 

Response 6. We Agree. We added the following statement to conflicts of interest, “Dr. Bussell is the primary investigator of this paper and has trained the physical therapy staff at NTC. NTC researchers are currently working on validating the NVI as well.” (page 11, conflicts of interest-paragraph 7, line 424-425)

Reviewer 2 Report

Comments and Suggestions for Authors Dear Editor,   Thank you for the opportunity to review this manuscript.   General comments: -Please fix all typographical errors throughout the manuscript.  -Revision of tenses would be recommended.   Abstract:

-Please paraphrase this sentence: "Researchers were able to retrospectively audit patient data collected from 1/2022-9/2024", " Researchers were able"....

-Please define abbreviations and acronyms at their first appearance in the text.   Introduction The introduction is well written, however, I would recommend to expand "the biological mechanism of INF® Therapy in improving microvascular function"    Methods: -Please define the criteria for “successful” completion of therapy (e.g., number of sessions). -Sample Size and Demographics  -Please justify the small sample size and discuss potential issues (in the discussion section).  -Please address the imbalance in neuropathy subgroups;  -Please describe how missing data were handled in the analysis.     Results: -Please include confidence intervals for all reported p-values; -Please also consider adding a table summarizing demographic variables or including demographic variables into the existing table (1);   Discussion: -Please provide a more thorough comparison with some prior studies using INF® therapy alone.  -Please elaborate on discussing potential mechanisms

Author Response

REVIEWER 2:

Comment 1. General comments: -Please fix all typographical errors throughout the manuscript.  -Revision of tenses would be recommended.  

Response 1. Thank you for your feedback on our manuscript submission. We agree and have fixed the tenses throughout the manuscript.

Comment 2. Please paraphrase this sentence: "Researchers were able to retrospectively audit patient data collected from 1/2022-9/2024", " Researchers were able"....

Response 2. Completed, we decided to remove the sentence, as it seemed redundant to the sentence prior.

Comment 3. -Please define abbreviations and acronyms at their first appearance in the text.   Introduction The introduction is well written, however, I would recommend to expand "the biological mechanism of INF® Therapy in improving microvascular function"    

Response 3. Thank you for correcting us. Completed. We corrected, “neurovascular index (NVI)” (Page 1, abstract- paragraph 1, line 16). We also added the following in the introduction section: “While these mechanical interventions are accepted for symptomatic relief and functional improvement, there is still a lack for a curative neuropathic treatment” (Page 2, introduction- paragraph 3, line 84-86). We also added, “The mechanical intervention used in this study was INF® therapy, which is a manual therapy treatment provided by a trained licensed physical therapist which aims to guide blood flow into the neural fascicle, improve endoneurial capillary circulation and ultimately reverse ischemia” (Page 3, introduction- paragraph 1, line 98-101). We also added more about INF therapy in the discussion when explaining the physiology we believe to be happening behind it.

Comment 4. Methods: -Please define the criteria for “successful” completion of therapy (e.g., number of sessions).

Response 4. We agree to this comment and have made the following addition. “Post NVI assessments they were given INF® Therapy treatments ranging from 4-13 visits.” (Page 3, Materials and methods- paragraph 2, line 131-133)

Comment 5.-Sample Size and Demographics  -Please justify the small sample size and discuss potential issues (in the discussion section). 

Response 5. Thank you for this comment, we agree. We have added the following: “Our small sample size is primarily due to our strict inclusion criteria, specifically, not including and subjects who left any questions blank. Investigators did not want to risk false data by assuming subjects may have meant 0, or no pain, on the form by not answering. Due to this exclusion criteria alone, 80 subjects were excluded. For our future studies, we hope to emphasize to our subjects to fill out our questionnaire in its entirety and emphasize the ability to ask questions if they are unsure, rather than leaving it blank.” (Page 10, Discussion-limitations- paragraph 2, line 378-384)

Comment 6. Please address the imbalance in neuropathy subgroups;  -Please describe how missing data were handled in the analysis.    

Response 6. We agree. We have added the following “. All patients regardless of neuropathy severity, or type of neuropathy, comorbidities, or length of symptomatic suffering were considered. Patients were excluded if they received only orthopedic treatment, were currently receiving their first INF® Therapy treatment, or had incomplete/missing data. Patients included in our study were asked to continue their daily lives as normal (meaning continue with their prescribed medication, and lifestyle).” (Page 3, Materials and methods- paragraph 2, line 123-128)

Comment 7. Results: -Please include confidence intervals for all reported p-values; -Please also consider adding a table summarizing demographic variables or including demographic variables into the existing table (1)

Response 7. Agreed and completed. We have added the confidence intervals for all P-values and updated the tables.

Comment 8. Discussion: -Please provide a more thorough comparison with some prior studies using INF® therapy alone.  -Please elaborate on discussing potential mechanisms

Response 8. Agreed and completed. We have added and edited the following “As expressed in the study by Alshahrani et. al., during the first hold of INF® Therapy nutrient vessels are stretched which subsequently enlarges the opening at the junction of the artery and bridging nutrient vessel [22]. These openings essentially create increased epifascial vascular pressure intending for a pressurized blood flow to cross into the endoneurial capillaries. However, due to a strong perineurium and increased endoneurial edema a secondary hold is necessary to drive blood flow into the endoneurial capillaries. This hold involves a mild stretch to allow for circulation to be biased past the perineurium and into these capillaries. Finally, a third hold is used to create a vacuum within the vasculature to pull microvasculature circulation into the intraneural vascular beds [23]. If this extra-capillary vascular pressure, exceeds the resistance of potentially closed neural capillaries, during the three holds of INF® Therapy, then red blood cells may be compressed. Studies show red blood cells that are forced through simulated tight capillary lumens secrete adenoside triphosphate (ATP) [24]. ATP then induces nitric oxide from capillary endothelial cells [25] facilitating endothelial vasodilation. Theoretically, inducing endothelial vasodilation should enhance blood flow in these areas and improve nerve function through the enhancement of the concentration of Na/K ATPase [22]. We hypothesize that, though there are few studies that have investigated this technique, this mechanism of INF® Therapy accounts for the significant changes seen in patient’s PQAS responses before and after treatment series. Given these results, we still believe that more research is needed to confirm this proposed mechanism, as well as how INF® Therapy can increase neural circulation and alleviate symptoms caused by ischemia-inducing conditions.” (Page 9-10, Discussion- paragraph 5, line 341-362). We also added more comparison with our prior studies, “The pain quality factors that were found to be significant were also found in the small nerve fiber symptoms, involving multiple symptoms such as burning, shooting, and numbness. This can indicate that INF® Therapy can affect small nerve fibers and alleviate conditions that restrict endoneurial blood flow. In this study, 18 out of the 20 domains were found significant, and although this study has a larger sample size (n=73), eliminating high inflammatory foods was not utilized in the Sahba et.al. study. However, the study did illuminate a placebo effect observed in the reduction of pain symptoms observed in both the INF and sham groups that they found to be consistent with the formation of a ‘sustained partnership’ between the healthcare provider and patient [21]. Thus, we believe that further research should be done exploring not only the importance of lifestyle modification in conjunction with INF® Therapy, but also including the impact that a provider-patient relationship formed through the administration of INF® Therapy can have on pain symptoms.” (Page 8, Discussion- paragraph 2, line 279-290). We also added: “These results share similarities with the study produced by Alshahrani et. al., where significance was seen between pre and posttreatment measurements of the modified Total Neuropathy Scale, Sensory Organization Test, and limits of stability test. These tests were important to test whether this therapy decreased symptoms of neuropathy, improved static balance, dynamic balance, and movement velocity.” (Page 8, Discussion- paragraph 3, line 300-305)

Reviewer 3 Report

Comments and Suggestions for Authors

Since the study is retrospective, there is a high possibility of bias in patient selection. Without a randomized controlled trial (RCT), it becomes difficult to separate the treatment effect from external factors.
There is no control group to compare the treatment effect in the study. This makes it difficult to determine whether the observed changes are due to INF Therapy or dietary changes.
The study does not specify whether the patients received other treatments simultaneously. Factors such as other medications used, lifestyle changes may mask or alter the treatment effect.
The specific criteria for which patients were included in the study were not clearly defined. For example, information such as neuropathy severity and comorbidities should be provided in more detail.

A power analysis was not performed to determine the statistical power of the study. It is not known whether a sample size of 73 people is sufficient.

No specific validation was performed on the sensitivity of the Pain Quality Assessment Scale (PQAS) scale used. In addition, the Neurovascular Index (NVI) method is a commercial method and it is unclear how reliable it is with objective scientific evaluations.

The study used Paired t-Test for the data, but did not test whether the data were normally distributed. If the distribution is not normal, non-parametric tests such as the Wilcoxon Signed-Rank Test should be used.

The mechanism of INF Therapy has been explained theoretically, but the biological evidence underlying it is limited. More data should be provided, especially on how the therapy affects microcirculation.

The study states that patients' symptoms improved after avoiding certain foods, but the underlying mechanisms are not clear.

Comments on the Quality of English Language

minor editing 

Author Response

REVIEWER 3:

Comment 1. Since the study is retrospective, there is a high possibility of bias in patient selection. Without a randomized controlled trial (RCT), it becomes difficult to separate the treatment effect from external factors.

Response 1. Thank you for your response to our paper and your knowledge on how to make our paper better as well as future studies. We agree a RCT is always best. Our hope with this study was to look at our past and analyze our treatment model looking retrospectively at all patients entering our clinic with neuropathy symptoms. We did not pick specific patients to be included, we have taken your advice and stated this in our study. “All patients regardless of neuropathy severity, or type of neuropathy, comorbidities, or length of symptomatic suffering were considered. Patients were excluded if they received only orthopedic treatment, were currently receiving their first INF® Therapy treatment, or had incomplete/missing data.” (Page 3, under Materials and methods- paragraph 2, lines 123-127)

Comment 2. There is no control group to compare the treatment effect in the study. This makes it difficult to determine whether the observed changes are due to INF Therapy or dietary changes.
The study does not specify whether the patients received other treatments simultaneously. Factors such as other medications used, lifestyle changes may mask or alter the treatment effect.
The specific criteria for which patients were included in the study were not clearly defined. For example, information such as neuropathy severity and comorbidities should be provided in more detail.

Response 2. We agree with your response. We have made the following changes. We have added “Patients included in our study were asked to continue their daily lives as normal (meaning continue with their prescribed medication, and lifestyle). Patients that began our program were given 2-4 NVI tests to assess for possible food allergens and were asked to abstain from these foods. We did not monitor their compliance with this. Post NVI assessments they were given INF® Therapy treatments ranging from 4-13 visits. Once INF® Therapy treatments were completed patients were given another PQAS form to complete to assess changes in their neuropathic pain.” (Page 3, under Materials and methods- paragraph 2, lines 127-133)

Comment 3. A power analysis was not performed to determine the statistical power of the study. It is not known whether a sample size of 73 people is sufficient.

Response 3. Thank you for this observation, we agree, and added it into our study (it was completed, just not added). “The power analysis was added, and the sample of 73 for variety of effect sizes gave power more that 85%. We added the following statement: “A power analysis was performed for a sample size of 73 subjects giving us an effect size of 0.85 at a level of significance α=0.05.” (Page 5, Materials and Methods-statistical analysis- paragraph 2, lines 200-201)

Comment 4. No specific validation was performed on the sensitivity of the Pain Quality Assessment Scale (PQAS) scale used. In addition, the Neurovascular Index (NVI) method is a commercial method and it is unclear how reliable it is with objective scientific evaluations.

Response 4. Thank you for your comment, we agree. We added the power analysis to the study to address this. Sensitivity (statistical power) in a paired t-test refers to the probability of correctly detecting a true effect when it exists. It is often assessed using power analysis. If the power is ≥ 0.8, the test is considered sufficiently sensitive. The power calculation was done, and it was more than 80%. This statement was added: “The power analysis was added, and the sample of 73 for variety of effect sizes gave power more that 85%. We added the following statement: “A power analysis was performed for a sample size of 73 subjects giving us an effect size of 0.85 at a level of significance α=0.05.” (Page 5, Materials and Methods-statistical analysis- paragraph 2, lines 200-201)

Comment 5. The study used Paired t-Test for the data, but did not test whether the data were normally distributed. If the distribution is not normal, non-parametric tests such as the Wilcoxon Signed-Rank Test should be used.

Response 5. Thank you for asking. It was very interesting that the normality test showed that most cases were normal, then the T-test would have been appropriate. However, in the few cases where normality was violated, the Wilcoxon Signed-Rank test would have been the safer choice. But the results were not very different in terms of being significant. We edited the following statement, “The paired t-test was used to compare pre- and post- variables within the group as most cases showed data was distributed normally, Wilcoxon Signed-Rank test was also run, results were not different in terms of significance.” (Page 5, Materials and Methods-statistical analysis- paragraph 2, lines 201-204)

Comment 6. The mechanism of INF Therapy has been explained theoretically, but the biological evidence underlying it is limited. More data should be provided, especially on how the therapy affects microcirculation.

Response 6. We agree with this comment, and we have added more data from previous papers, however our therapy is very new, and we are hoping to continue future studies to analyze the underlining mechanism in greater detail biologically. We added and edited the following: “As expressed in the study by Alshahrani et. al., during the first hold of INF® Therapy nutrient vessels are stretched which subsequently enlarges the opening at the junction of the artery and bridging nutrient vessel [22]. These openings essentially create increased epifascial vascular pressure intending for a pressurized blood flow to cross into the endoneurial capillaries. However, due to a strong perineurium and increased endoneurial edema a secondary hold is necessary to drive blood flow into the endoneurial capillaries. This hold involves a mild stretch to allow for circulation to be biased past the perineurium and into these capillaries. Finally, a third hold is used to create a vacuum within the vasculature to pull microvasculature circulation into the intraneural vascular beds [23]. If this extra-capillary vascular pressure, exceeds the resistance of potentially closed neural capillaries, during the three holds of INF® Therapy, then red blood cells may be compressed. Studies show red blood cells that are forced through simulated tight capillary lumens secrete adenoside triphosphate (ATP) [24]. ATP then induces nitric oxide from capillary endothelial cells [25] facilitating endothelial vasodilation. Theoretically, inducing endothelial vasodilation should enhance blood flow in these areas and improve nerve function through the enhancement of the concentration of Na/K ATPase [22]. We hypothesize that, though there are few studies that have investigated this technique, this mechanism of INF® Therapy accounts for the significant changes seen in patient’s PQAS responses before and after treatment series. Given these results, we still believe that more research is needed to confirm this proposed mechanism, as well as how INF® Therapy can increase neural circulation and alleviate symptoms caused by ischemia-inducing conditions.” (Page 9-10, Discussion- paragraph 5, line 341-362)

Comment 7. The study states that patients' symptoms improved after avoiding certain foods, but the underlying mechanisms are not clear.

Response 7.  We agree. We are working on another research study to examine what may be happening physiologically. In our discussion section we aimed to address what we believe is happening when ingesting high inflammatory foods and cited those articles who found similar results. Due to the retrospective nature of the study, we can merely suggest what we observed with those who avoided inflammatory foods. We hope to continue researching this further to really understand what is happening physiologically.

Round 2

Reviewer 1 Report

Comments and Suggestions for Authors

Many of the things I pointed out before were improved. It was also clarified. The author's conflict, which was the most worrisome, was also well explained. Therefore, there is no more to point out.

Reviewer 3 Report

Comments and Suggestions for Authors

It can be accepted